# Changes in Smoking, Alcohol Consumption, and the Risk of Thyroid Cancer: A Population-Based Korean Cohort Study

**DOI:** 10.3390/cancers13102343

**Published:** 2021-05-12

**Authors:** Yohwan Yeo, Kyungdo Han, Dong-Wook Shin, Dahye Kim, Su-Min Jeong, Sohyun Chun, In-Young Choi, Keun-Hye Jeon, Tae-Hyuk Kim

**Affiliations:** 1Department of Family Medicine & Supportive Care Center, Samsung Medical Center, Sungkyunkwan University School of Medicine, Seoul 06351, Korea; yohwan.yeo@samsung.com (Y.Y.); sm2021.jeong@samsung.com (S.-M.J.); sohyun.chun@samsung.com (S.C.); inyoungb.choi@samsung.com (I.-Y.C.); 2Department of Statistics and Actuarial Science, Soongsil University, Seoul 06978, Korea; 3Department of Clinical Research Design & Evaluation, Samsung Advanced Institute for Health Science & Technology (SAIHST), Sungkyunkwan University, Seoul 06351, Korea; 4Department of Digital Health (SAIHST), Sungkyunkwan University, Seoul 06351, Korea; 5Department of Medical Statistics, The Catholic University of Korea, Seoul 06591, Korea; dhkim373@daewoong.co.kr; 6International Healthcare Center, Samsung Medical Center, Seoul 06351, Korea; 7CHA Gumi Medical Center, Department of Family Medicine, Gumi 39295, Korea; kh1228@chamc.co.kr; 8Thyroid Center, Samsung Medical Center, Division of Endocrinology and Metabolism, Department of Internal Medicine, Sungkyunkwan University School of Medicine, Seoul 06351, Korea; taehyukmd.kim@samsung.com

**Keywords:** smoking, alcohol intake, lifestyle change, thyroid cancer, cohort study

## Abstract

**Simple Summary:**

The inverse association between smoking, alcohol intake, and thyroid cancer has been suggested by observational studies. From the representative data in Korea, we identified the epidemiologic evidence to elucidate the true effect between smoking, alcohol intake, and thyroid cancer incidence by exploring the effect of changes in smoking and alcohol consumption habits.

**Abstract:**

To elucidate potential causality between smoking and alcohol intake on thyroid cancer incidence, we explored the effect of changes in smoking and alcohol consumption habits. From the Korean National Health Insurance database, we identified 4,430,070 individuals who participated in the national health screening program in 2009 and 2011. The level of smoking and alcohol consumption was measured twice, once in 2009 and again in 2011. The risk of thyroid cancer according to their changes was estimated using the Cox proportional hazard model. During the mean follow-up period of 6.32 ± 0.72 years, 29,447 individuals were diagnosed with thyroid cancer. Compared to those who sustained not smoking, non-smokers who initiated smoking to light (adjusted hazard ratio (aHR) 0.96, 95% confidence interval (CI) 0.81–1.15), moderate (aHR 0.90, 95% CI 0.78–1.04), and heavy level (aHR 0.81, 95% CI 0.69–0.96) had a decreased risk of thyroid cancer. Heavy smokers who quit smoking had an increased risk of thyroid cancer (aHR 1.23, 95% CI 1.06–1.42) compared to those who sustained heavy smoking. Change in drinking status was not significantly associated with thyroid cancer risk compared to drinking at the same level, although a non-significant trend of increased risk was noted in quitters. Participants who initiated both smoking and drinking (HR 0.80, 95% CI 0.69–0.93) had a lower risk of thyroid cancer compared with those who continued not to smoke and drink. Our findings provide further evidence that smoking, and possibly alcohol consumption, would have true protective effects on the development of thyroid cancer.

## 1. Introduction

Previous studies have found that the risk of thyroid cancer incidence was inversely associated with smoking [1,2,3,4,5,6] and alcohol consumption [5,6,7,8,9]. In our recent longitudinal study from a large, representative population (≈10 million people) in Korea where the incidence of thyroid cancer is relatively high, we also reported that smoking and alcohol drinking are independent factors that are inversely associated with thyroid cancer incidence [6].

However, the issues of overdiagnosis on thyroid cancer incidence have complicated the interpretation of this inverse association between smoking, alcohol intake, and thyroid cancer, because those who opt for thyroid cancer screening are also likely to have better health behaviors, i.e., low prevalence of smoking and alcohol consumption. Improvements in diagnostic tools and increased accessibility to medical services might cause an increased detection of small and asymptomatic lesions [10]. This led to the increase in incidence [11] without any change in thyroid cancer-related mortality [12]. In Korea, the increased use of private health screening services, which sometimes includes screening for thyroid cancer, have been related with the overdiagnosis of thyroid cancer [13]. The situation was similar in other developed countries: between 1998 and 2008 in the United States, about 80% (228,000 cases) of small and asymptomatic thyroid cancer cases were estimated to be the results of overdiagnosis, followed by about 70% of cases in Italy, France, and Australia, and 50% of cases in Nordic countries and England [10].

Persons who participate in private health screening programs are mostly high-income people who generally smoke less and drink less [13,14,15]. Although our previous study did not find significant effect modification by income status in the association between smoking or alcohol consumption and thyroid cancer, suggesting a true effect rather than confounding [6], there is also concern that the inverse association is due to thyroid screening practices among people who are more engaged in health screening and better health behavior.

In this context, we thought that one way to elucidate the true association between smoking, alcohol intake and thyroid cancer risk in an observational study was to explore the effect of changes in smoking and alcohol consumption habits. For example, if smoking or drinking is protective, quitting might lead to increased thyroid cancer incidence, and initiating would lead to decreased thyroid cancer incidence. Therefore, we sought to investigate the association between changes in smoking and alcohol intake and thyroid cancer risk using representative data in Korea.

## 2. Methods

### 2.1. Study Design and Population

We conducted a retrospective cohort study based on the Korean National Health Insurance (KNHI) database (access on the date of 3 May 2021). This study includes data on inpatient visits, outpatient visits, procedures, and prescription medications covered by the KNHI, a mandatory universal public health insurance system that covers the entire Korean population except for Medicaid beneficiaries in the lowest income bracket (approximately 3% of the population).

The KNHI database includes data on qualification (e.g., age, sex, income, region, and type of eligibility), claims (e.g., general information on specifications; consultation statements; diagnosis statements defined by the International Classification of Diseases, 10th revision (CD-10); and prescription statements, and health check-ups [16,17]. The KNHI provides biennial cardiovascular health screenings to subjects who are aged 40 and above or employees; the screening includes self-administered health questionnaires on lifestyle factors (smoking, alcohol consumption, exercise, etc.), anthropometric measurements by clinical staff (height, weight, blood pressure, etc.), and laboratory tests after overnight fasting [18].

From the KNHI database of the entire Korean population, we collected participants ≥40 years old who had undergone two consecutive national health examinations in both 2009–2010 (first) and 2011–2012 (second) to determine changes in smoking and alcohol intake. Among the 4,961,441 individuals who underwent the two health examinations during these years, participants with a history of any type of cancer (registration in cancer registry with ICD-10 C-codes) (*N* = 141,566) or cardiovascular disease (*N* = 65,108) prior to the second health examination were excluded. We applied a one-year lag time by excluding participants who were diagnosed with any type of cancer (*N* = 50,605) or with cardiovascular disease (*N* = 14,054) or who died (*N* = 7820) within 1 year after the second health examination period in order to exclude persons who quit or decrease their smoking or drinking due to their worsened medical condition (sick quitter effect). Those with missing information in key variables (*N* = 252,218) were also excluded. Finally, we included a total of 4,430,070 individuals (Figure 1). This study was approved by the institutional review board (IRB) of Samsung Medical Center (IRB File No. SMC 2019–01-024).

### 2.2. Smoking and Alcohol Consumption

Smoking behaviors: Information on smoking status and changes in smoking habits was obtained from the self-administered questionnaire of the biennial national health examination in the KNHI. They were classified into the categories of “none”, “past”, or “current” smoking based on how they responded to the baseline survey. Participants who answered ‘yes, and I am currently smoking’ to the question were classified as current smokers. Others who answered in the affirmative to, ‘Have you ever smoked at least 100 cigarettes (5 packs) in your entire life?’ at each health examination period were classified as ever-smokers.

Data on the amount of smoking were also gathered as an average amount of daily smoking among ever-smokers and current smokers. They were asked about the average number of cigarettes smoked per day and duration of smoking in years. According to the number of cigarettes per day in the first examination (2009–2010), currently, smoking study participants were categorized into three groups: (1) heavy smokers (≥20 cigarettes per day), (2) moderate smokers (10–19 cigarettes per day), and (3) light smokers (<10 cigarettes per day).

Alcohol intake: Alcohol intake was categorized based on the nature of alcohol consumption per week (“none”, “light”, “moderate”, or “heavy”). Participants reported how many times per week they usually drank as well as how many glasses they drank per setting, and the weekly amount of alcohol consumption was calculated by multiplying the two. If subjects reported drinking less than 15 g/day on average, they were classified as “light” drinkers, while those who drank more than 15 g/day but less than 30 g/day were “moderate” drinkers, and those who drank more than 30 g/day were “heavy” drinkers.

### 2.3. Potential Confounders

Several variables that might be associated with thyroid cancer were regarded as confounders, including sex, body mass index (BMI) [19], regular exercise [1,20], monthly income [21], and chronic illnesses including diabetes [22,23] and hyperlipidemia [24,25]. BMI was calculated by weight/height squared (kg/m^2^) and was divided into five groups (<18.5, 18.5–23, 23–25, 25–30, and ≥30 kg/m^2^). Regular exercise was defined as engaging in physical activity of a moderate-to-vigorous intensity for three or more days per week. Income level was assessed by monthly insurance premium, as insurance contribution is determined by income level rather than health risk in Korea.

The presence of comorbidities was defined by diagnostic codes and the prescription of relevant medications or by health check-up results. Diabetes was defined as ICD-10 codes E10 through E14 with at least one prescription of an antidiabetic medication, or a fasting glucose level of 126 mg/dL or more, and hyperlipidemia was defined as ICD-10 code E78 with at least one prescription of a lipid-lowering agent, or a total cholesterol level of 240 mg/dL or more.

### 2.4. Outcome Measurement

The incidence of thyroid cancer was defined based on diagnosis codes for thyroid cancer (i.e., C73) registered at least 1 year after the second health examination with inclusion in a special copayment reduction program for critical illness [26]. In Korea, virtually all people apply for this program if they are diagnosed with cancer because a 5% copayment applies for the work-up and treatment for cancer (vs. 20–30% for other common diseases). For this reason, cancer incidence in Korea is rarely omitted from this claims database and is sufficiently reliable. Participants were followed up from the date of entry to the date of thyroid cancer diagnosis, the end of follow-up, or the end date of the study 31st December 2018, whichever came first.

### 2.5. Statistical Analysis

Descriptive statistics were used to determine the basic characteristics of the study population. Multivariate Cox proportional hazards regression analyses were employed to determine the hazard ratio (HR) for thyroid cancer incidence further categorized by smoking and alcohol consumption status. The covariates included in the regression models were non-adjusted (model 1), age and sex (model 2), and those in model 2 plus BMI, regular exercise, monthly income, diabetes mellitus, and dyslipidemia (model 3). The analyses were made with two reference groups: one with sustained non-smoking or non-drinking as the reference, and the other with the smoking or drinking status from the first examination as the reference. Potential interactions between smoking and drinking were examined by combining smoking and alcohol consumption status (4 × 4 design). The proportional hazards assumption was checked by using the Schoenfeld residuals with the logarithm of the cumulative hazards function estimated by Kaplan–Meier method. All analyses were performed using SAS (version 9.4; SAS Institute, Cary, NC, USA), and the results with p-values of less than 0.05 were considered as statistically significant.

## 3. Results

During the mean follow-up period of 6.32 ± 0.72 years, 29,447 individuals were diagnosed with thyroid cancer. The mean time to thyroid cancer incidence was 2.66 ± 1.88 years. Baseline characteristics of the subjects are shown in Table 1. Thyroid cancer patients were a little younger (53.4 ± 8.4 years in the thyroid cancer group vs. 55.6 ± 10.0 years in the non-thyroid cancer group) and more were female (77.0% in the thyroid cancer group vs. 48.8% in the non-cancer group). Additionally, there were fewer individuals who reported themselves as moderate to heavy smokers (6.9% vs. 16.8%) and mild to heavy drinkers (30.0% vs. 42.2%) in the thyroid cancer group compared to the non-thyroid cancer group (Table 1).

### 3.1. Risk of Thyroid Cancer by Changes in Cigarette Smoking

Compared to the sustained non-smokers, sustained light smokers (adjusted hazard ratio (aHR) 0.77, 95% confidence interval (CI) 0.65–0.92), sustained moderate smokers (aHR 0.75, 95% CI 0.68–0.82), and sustained heavy smokers (aHR 0.71, 95% CI 0.66–0.77) had lower risk of thyroid cancer, respectively. In addition, those who initiated smoking in the light (aHR 0.96. 0.81–1.15), moderate (aHR 0.90, 95% CI 0.78–1.04), or heavy levels (aHR 0.81, 95% CI 0.69–0.96) was associated with a decreased incidence of thyroid cancer (Table 2).

Among heavy smokers at the first examination, quitters had an increased risk of thyroid cancer (HR 1.23, 95% CI 1.06–1.42) and those who reduced to light smoking also showed a similar trend, although it was not significant (aHR 1.28, 95% CI 0.85–1.95) compared to those who sustained heavy smoking. For those who were mild or moderate smokers at the first examination, quitting, reducing, or increasing smoking level were not significantly associated with thyroid cancer risk compared to those who continued smoking at the same level.

### 3.2. Risk of Thyroid Cancer by Changes in Alcohol Consumption

Table 3 shows the risk of thyroid cancer according to alcohol intake and their changes at the follow-up screening. Compared to those who sustained non-drinking, sustained drinkers at all levels showed a lower risk of thyroid cancer: light sustainers (HR 0.90 95% CI 0.87–0.94), moderate sustainers (HR 0.88, 95% CI 0.81–0.96), and heavy sustainers (HR 0.90, 95% CI 0.82–1.00). In the analysis by stratification of drinking intensity at the first examination, there were no significant results regarding the change in alcohol intake.

### 3.3. Risk of Thyroid Cancer by Joint Effect of Changes in Smoking and Alcohol Consumption

Compared to the individuals who had never smoked or drank, sustainers of both smoking and drinking had an attenuated lower risk of thyroid cancer: smoking sustainers (aHR 0.83, 95% CI 0.77–0.90), drinking sustainers (aHR 0.89, 95% CI 0.86–0.93), and sustainers of both (aHR 0.72, 95% CI 0.68–0.77) (Table 4).

When stratifying by smoking and alcohol intake at the first examination, participants who initiated both smoking and drinking (aHR 0.80, 95% CI 0.69–0.93) had a lower risk of thyroid cancer compared with those who continued not to smoke and drink. Among persons who had smoked and drank at the first examination, smoking quitters (aHR 1.20, 95% CI 1.07–1.36) or drinking quitters (aHR 1.20, 95% CI 1.07–1.34) experienced a higher risk of thyroid cancer compared with those who sustained both smoking and drinking. A higher risk was found in persons who quit both smoking and drinking (aHR 1.10, 95% CI 0.94–1.29), but it was not statistically significant.

## 4. Discussion

To the best of our knowledge, this is the first study to date that has investigated the association between changes in smoking and alcohol intake, and thyroid cancer incidence. In addition to the decreased thyroid cancer risk in smokers and drinkers found in previous studies, our study showed that changes in smoking status may change thyroid cancer risk, providing further evidence of a protective effect of smoking on thyroid cancer.

### 4.1. Smoking and Thyroid Cancer

This study confirmed that smoking was associated with a decreased risk of thyroid cancer with a dose–response relationship according to smoking intensity. Heavy smokers showed the lowest risk of thyroid cancer in this study. This is consistent with the findings of other recent cohort studies and meta-analyses, which showed a dose–response relationship with smoking intensity [1,5], duration [5], or pack-years [2,5]. The pathophysiological mechanisms for the association between smoking and thyroid cancer included thyroid-stimulating hormone (TSH) suppression [27,28], BMI reduction [29], the anti-estrogenic effect of smoking [30,31], and the protective effect of nicotine [32].

### 4.2. Changes in Smoking and Thyroid Cancer

Our study showed that those who initiated smoking had a lower risk than sustained non-smokers, and those who quit from heavy smoking showed a higher risk than sustained heavy smokers. As our participants are limited to individuals who underwent two serial health examinations, there is no chance that this smoking habit change is associated with screening behavior. Therefore, our study strongly suggests a true association between smoking and thyroid cancer, rather than overdiagnosis of thyroid cancer in non-smokers, who have better health behaviors.

Some studies have reported inconsistent results regarding the dose–response relationship by year since quitting: one study found a weaker reduction in risk with greater number of years since quitting [5], while the other did not [3]. In our study, which used a 2-year interval, however, we could not assess the number of years that are needed following smoking cessation to eliminate the protective effect of smoking.

### 4.3. Alcohol Intake and Thyroid Cancer

Our study definitely showed a lower thyroid cancer risk by alcohol consumption, which is consistent with previous studies [5,6,7,8,9]. In this study, sustained alcohol consumption at light to heavy levels was associated with an approximately 10% lower risk of thyroid cancer, without a clear dose–response relationship. The decrease in TSH level with alcohol consumption [28,33] and the direct toxic effect of alcohol on thyroid cells [34,35] have been suggested as potential mechanisms for the association between alcohol consumption and thyroid cancer risk.

### 4.4. Changes in Alcohol Consumption and Thyroid Cancer

However, in our study, changes in drinking status were not significantly associated with thyroid cancer risk compared to drinking at the same levels, although a non-significant trend for increased risk was noted in quitters (e.g., aHR = 1.04 (95% CI 0.99–1.10) for mild drinking quitters, and aHR = 1.10 (95%CI 0.88–1.38) for heavy drinking quitters). The reasons for this non-significant association would be as follows. First, the magnitude of the inverse association in alcohol intake was relatively smaller than that of smoking. For example, sustained heavy drinkers had about a 10% lower risk compared with sustained non-drinkers, while sustained heavy smokers had a 30% lower risk than non-smokers. Therefore, the changes in thyroid cancer risk resulting from a change in drinking status would be too small to detect statistically. Second, drinking patterns are usually more irregular than those of smoking. For example, some people drink more often during some seasons (e.g., holidays), and some do not drink at all in other seasons when they are busy, while smoking amount is relatively constant regardless of timing [36]. Therefore, the difficulty in the assessment of an individual’s drinking habits might partly lead to the null association seen in this study between the change in alcohol consumption and thyroid cancer risk.

### 4.5. Joint Effect of Changes in Smoking and Alcohol Intake

As smoking and drinking habits are often correlated and would have collinearity issues, we estimated their joint effect. In this study, when comparing persons who continued non-smoking and non-drinking to sustainers for both smoking and drinking, the sustainers had a pronounced lower risk of thyroid cancer than those who either sustained smoking or drinking only. Our previous study [6], as well as a pooled-analysis by Kitahara et al. [5], found a sub-multiplicative interaction between smoking and alcohol intake on thyroid cancer risk. This effect was also consistent, as we found that quitting smoking or alcohol was associated with a 20% higher risk of thyroid cancer than those who sustained both smoking and drinking. The lack of association in those who quit both smoking and drinking may be attributable to the small sample size of those individuals (only 4.1% quit both smoking and drinking at the 2-year follow-up).

### 4.6. Limitations

The limitations of this study are as follows: First, our data did not include information on the type of thyroid cancer, stage, or genomic biomarkers for thyroid cancer, even though almost all thyroid cancer cases (>99.0%) in Korea have been reported as papillary thyroid cancer [25]. Second, information on the individual’s participation in thyroid cancer screening was not available but could have been helpful in confirming whether this association is reliable despite the potential confounding effect of over-diagnosis due to thyroid cancer screening. Third, self-reported smoking habits and alcohol consumption might not be accurate and are likely to be underestimated, which leads to a less prominent inverse association. Fourth, we did not have information on TSH levels, which could be a potential confounder or mediator. Fifth, while smoking and drinking behaviors can constantly change, we measured these behaviors only at two timepoints and did not reflect changes that were made in the rest of the follow-up period. Finally, our follow-up period (mean = 6.3 years) was relatively short. However, a previous study suggested that the inverse association between smoking/alcohol consumption and thyroid cancer incidence did not differ significantly by follow-up time (<5 years, 5–10 years, and ≥10 years) [5]. Therefore, we assume that our follow-up time would not have affected the results.

## 5. Conclusions

We confirmed that an inverse association exists between smoking and alcohol intake and thyroid cancer. In addition, we also showed that those who initiated smoking showed a lower risk than non-smoking sustainers, and those who quit heavy smoking showed a higher risk than sustained heavy smoker, suggesting a true protective effect of smoking on thyroid cancer risk. Changes in alcohol consumption status were not significantly associated with thyroid cancer as changes were in smoking, probably due to the smaller effect of alcohol on thyroid cancer risk.

## Figures and Tables

**Figure 1 cancers-13-02343-f001:**
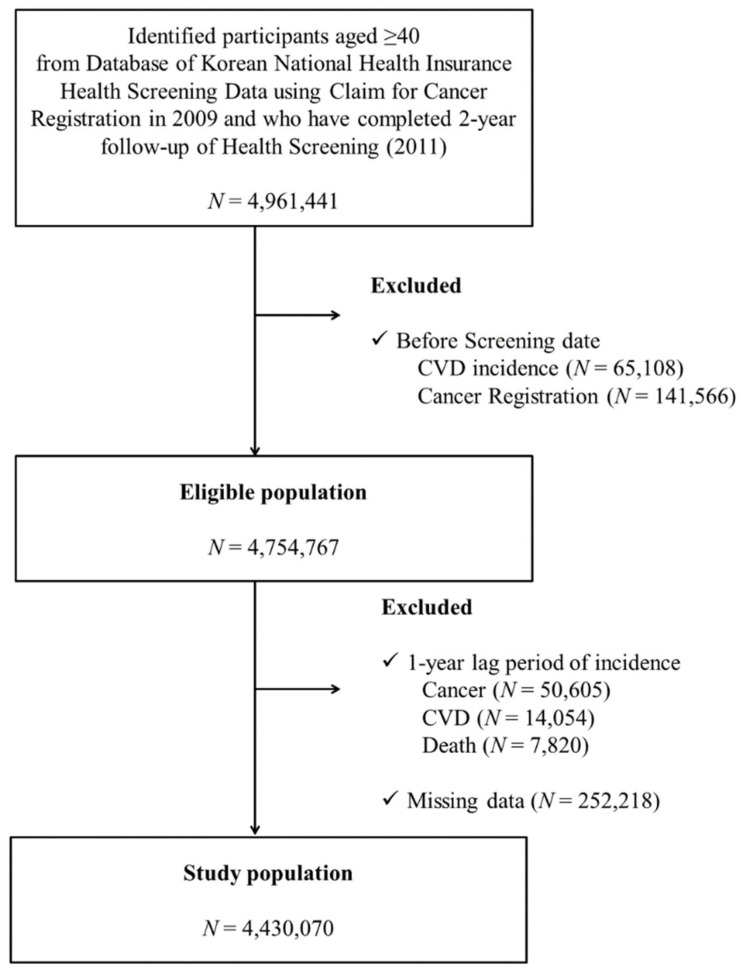
Flow chart of study participant enrollment.

**Table 1 cancers-13-02343-t001:** Baseline characteristics of thyroid cancer and non-thyroid cancer cases.

Characteristics	Total (*N* = 4,430,070)	Thyroid Cancer (*N* = 29,447)	Non-Cancer (*N* = 4,400,623)	*p*-Value
Age (year)	55.6 ± 10.0	53.4 ± 8.4	55.6 ± 10.0	<0.001
Sex (male) (*N*, %)	2,256,101 (50.9)	6757 (23.0)	2,249,344 (51.1)	<0.001
Body mass index (kg/m^2^)	24.0 ± 3.0	24.2 ± 3.1	24.0 ± 3.0	<0.001
Residence (urban area) (*N*, %)	1,986,395 (44.8)	14,312 (48.6)	1,972,083 (44.8)	<0.001
Smoking intensity (*N*, %)				<0.001
Non-smokers	3,597,067 (81.2)	27,063 (91.9)	3,570,004 (81.1)	
Light (<0.5 pack/day)	91,272 (2.2)	338 (1.2)	90,934 (2.1)	
Moderate (0.5–1 pack/day)	324,851 (7.3)	967 (3.3)	323,884 (7.4)	
Heavy (≥1 pack/day)	416,880 (9.4)	1079 (3.7)	415,801 (9.5)	
Smoking duration (year) (*N*, %)				<0.001
Non-smoker	2,818,291 (63.6)	24,395 (82.8)	2,793,896 (63.5)	
<5	64,215 (1.5)	304 (1.0)	63,911 (1.5)	
5–9	84,124 (1.9)	333 (1.1)	83,791 (1.9)	
10–19	353,904 (8.0)	1303 (4.4)	352,601 (8.0)	
20–29	603,648 (13.6)	1925 (6.5)	601,723 (13.7)	
≥30	505,888 (11.4)	1187 (4.0)	504,701 (11.5)	
Smoking history (pack-year) (*N*, %)				<0.001
Non-smoker	2,818,291 (63.6)	24,395 (82.8)	2,793,896 (63.5)	
<10	398,411 (9,0)	1562 (5.3)	396,849 (9.0)	
10–19	463,189 (10.5)	1459 (5.0)	461,730 (10.5)	
20–29	368,321 (8.3)	1060 (3.6)	367,261 (8.4)	
≥30	381,858 (8.6)	971 (3.3)	380,887 (8.7)	
Alcohol intake (*N*, %)				<0.001
None	2,561,862 (57.8)	20,601 (70.0)	2,541,261 (57.8)	
Mild (<15 g/day)	1,146,967 (25.9)	6354 (21.6)	1,140,613 (25.9)	
Moderate (15–30 g/day)	433,527 (9.8)	1553 (5.3)	431,974 (9.8)	
Heavy (≥30 g/day)	287,714 (6.5)	939 (3.2)	286,775 (6.5)	
Regular exercise (yes) (*N*, %)	971,781 (22.0)	6390 (21.7)	965,391 (21.9)	0.326
Income level (*N*, %)				
Q1	1,013,349 (22.9)	7265 (24.7)	1,006,084 (22.9)	
Q2	799,440 (18.1)	5636 (19.1)	793,804 (18.0)	
Q3	1,033,766 (23.3)	6330 (21.5)	1,027,436 (23.4)	
Q4	1,583,515 (35.7)	10,216 (34.7)	1,573,299 (35.8)	
Comorbidities (yes) (*N*, %)				
Diabetes mellitus, yes	530,288 (12.0)	2685 (9.1)	527,603 (12.0)	<0.001
Dyslipidemia, yes	1,593,542 (36.0)	10,017 (34.2)	1,583,525 (36.0)	<0.001

**Table 2 cancers-13-02343-t002:** Changes in smoking and thyroid cancer risk.

Smoking Intensity		Total	Incidence	Person-Years	Incidence Rate	Sustained Non-Smokers as the Reference			Smoking Status at the First Examination as the Reference		
At Initial	At 2-Year Follow-Up					Crude HR (95% CI)	Age and Sex Adjusted HR (95% CI)	Multivariate Adjusted * HR (95% CI)	Crude HR (95% CI)	Age and Sex Adjusted HR (95% CI)	Multivariate Adjusted * HR (95% CI)
None	None	3,414,941	26,400	21,623,575.19	1.22	1 (Reference)	1 (Reference)	1 (Reference)	1 (Reference)	1 (Reference)	1 (Reference)
	Light	26,539	128	166,011.36	0.77	0.63 (0.53, 0.75)	0.93 (0.78, 1.10)	0.96 (0.81, 1.15)	0.63 (0.53, 0.75)	0.92 (0.78, 1.10)	0.96 (0.81, 1.14)
	Moderate	51,311	189	322,074.03	0.59	0.48 (0.42, 0.55)	0.87 (0.75, 1.00)	0.90 (0.78, 1.04)	0.48 (0.42, 0.55)	0.86 (0.75, 0.996)	0.89 (0.77, 1.03)
	Heavy	48,927	145	306,819.47	0.47	0.39 (0.33, 0.45)	0.79 (0.67, 0.93)	0.81 (0.69, 0.96)	0.39 (0.33, 0.46)	0.78 (0.66, 0.92)	0.81 (0.69, 0.95)
Light	None	33,505	150	209,655.97	0.72	0.58 (0.50, 0.69)	0.82 (0.70, 0.97)	0.84 (0.72, 0.99)	1.23 (0.97, 1.56)	1.11 (0.87, 1.41)	1.09 (0.86, 1.39)
(<0.5 pack/day)	Light	33,710	122	209,883.87	0.58	0.47 (0.40, 0.57)	0.73 (0.61, 0.88)	0.77 (0.65, 0.92)	1 (Reference)	1 (Reference)	1 (Reference)
	Moderate	16,365	54	101,994.49	0.53	0.43 (0.33, 0.56)	0.75 (0.57, 0.98)	0.79 (0.60, 1.03)	0.91 (0.66, 1.26)	1.03 (0.75, 1.42)	1.06 (0.77, 1.46)
	Heavy	4317	11	26,862.08	0.41	0.33 (0.19, 0.60)	0.66 (0.37, 1.20)	0.70 (0.39, 1.27)	0.71 (0.38, 1.31)	0.93 (0.50, 1.72)	0.99 (0.53, 1.85)
Moderate	None	72,326	257	454,445.36	0.57	0.46 (0.41, 0.52)	0.85 (0.75, 0.96)	0.86 (0.76, 0.98)	1.24 (1.06, 1.43)	1.18 (1.01, 1.37)	1.11 (0.95, 1.29)
(0.5–1 pack/day)	Light	24,256	65	151,153.64	0.43	0.35 (0.28, 0.45)	0.63 (0.49, 0.80)	0.67 (0.52, 0.85)	0.94 (0.73, 1.22)	0.86 (0.67, 1.12)	0.86 (0.67, 1.12)
	Moderate	181,138	521	1,136,992.88	0.46	0.37 (0.34, 0.41)	0.71 (0.65, 0.78)	0.75 (0.68, 0.82)	1 (Reference)	1 (Reference)	1 (Reference)
	Heavy	53,944	137	337,421.79	0.41	0.33 (0.28, 0.39)	0.66 (0.56, 0.78)	0.69 (0.59, 0.82)	0.89 (0.73, 1.07)	0.93 (0.77, 1.12)	0.95 (0.79, 1.15)
Heavy	None	76,295	256	479,001.41	0.53	0.44 (0.39, 0.49)	0.92 (0.81, 1.04)	0.92 (0.81, 1.05)	1.32 (1.14, 1.52)	1.31 (1.14, 1.51)	1.23 (1.06, 1.42)
(≥1 pack/day)	Light	6767	23	42,107.32	0.55	0.45 (0.30, 0.67)	0.91 (0.61, 1.37)	0.95 (0.63, 1.44)	1.34 (0.89, 2.04)	1.28 (0.85, 1.94)	1.29 (0.85, 1.95)
	Moderate	76,037	203	475,687.6	0.43	0.35 (0.30, 0.40)	0.71 (0.62, 0.81)	0.74 (0.65, 0.86)	1.05 (0.90, 1.23)	1.02 (0.88, 1.20)	1.04 (0.89, 1.22)
	Heavy	309,692	786	1,939,574.2	0.41	0.33 (0.31, 0.36)	0.69 (0.64, 0.74)	0.71 (0.66, 0.77)	1 (Reference)	1 (Reference)	1 (Reference)

Incidence rate: per 1000 person-years. * Multivariate adjusted for: age, sex, regular exercise, monthly income, BMI, alcohol consumption, diabetes mellitus, and dyslipidemia. BMI, body mass index; HR, hazard ratio; CI, 95% confidence interval.

**Table 3 cancers-13-02343-t003:** Changes in alcohol intake and thyroid cancer risk.

Drinking Habit		Total	Incidence	Person-Years	Incidence Rate	Sustained Non-Drinkers as the Reference			Drinking Status at the First Examination as the Reference		
At Initial	At 2-Year Follow-Up					Crude HR (95% CI)	Age and Sex Adjusted HR (95% CI)	Multivariate Adjusted * HR (95% CI)	Crude HR (95% CI)	Age and Sex Adjusted HR (95% CI)	Multivariate Adjusted * HR (95% CI)
None	None	2,192,774	18,230	13,866,850.42	1.31	1 (Reference)	1 (Reference)	1 (Reference)	1 (Reference)	1 (Reference)	1 (Reference)
	Light	291,663	2096	1,844,051.74	1.14	0.86 (0.83, 0.90)	0.96 (0.92, 1.01)	0.97 (0.92, 1.01)	0.86 (0.83, 0.90)	0.96 (0.91, 1.00)	0.96 (0.92, 1.01)
	Moderate	41,871	184	263,993.07	0.70	0.53 (0.46, 0.61)	0.90 (0.78, 1.04)	0.91 (0.78, 1.05)	0.53 (0.46, 0.61)	0.90 (0.78, 1.04)	0.91 (0.78, 1.05)
	Heavy	23,916	103	149,949.06	0.69	0.52 (0.43, 0.63)	1.05 (0.87, 1.28)	1.07 (0.88, 1.31)	0.52 (0.43, 0.63)	1.06 (0.87, 1.29)	1.07 (0.88, 1.31)
Light	None	299,509	2073	1,891,864.16	1.10	0.83 (0.80, 0.87)	0.95 (0.91, 1.00	0.95 (0.91, 1.00)	1.28 (1.22, 1.35)	1.06 (0.998, 1.12)	1.04 (0.99, 1.10)
(<15 g/day)	Light	647,455	3499	4,093,232.18	0.85	0.65 (0.63, 0.67)	0.89 (0.86, 0.93)	0.90 (0.87, 0.94)	1 (Reference)	1 (Reference)	1 (Reference)
	Moderate	133,920	514	845,656.2	0.61	0.46 (0.42, 0.50)	0.86 (0.78, 0.94)	0.87 (0.80, 0.96)	0.71 (0.65, 0.78)	0.95 (0.87, 1.05)	0.97 (0.88, 1.06)
	Heavy	40,495	139	254,802.99	0.55	0.41 (0.35, 0.49)	0.85 (0.72, 1.00)	0.87 (0.74, 1.03)	0.64 (0.54, 0.76)	0.94 (0.79, 1.11)	0.96 (0.81, 1.14)
Moderate	None	43,044	192	270,033.78	0.71	0.54 (0.47, 0.62)	0.93 (0.81, 1.08)	0.92 (0.80, 1.07)	1.30 (1.11, 1.53)	1.08 (0.91, 1.28)	1.04 (0.87, 1.23)
(15–30 g/day)	Light	156,105	593	984,286.9	0.60	0.46 (0.42, 0.50)	0.85 (0.78, 0.93)	0.87 (0.80, 0.95)	1.11 (0.99, 1.24)	0.99 (0.89, 1.12)	0.99 (0.88, 1.11)
	Moderate	171,760	590	1,083,760.29	0.54	0.41 (0.38, 0.45)	0.86 (0.79, 0.93)	0.88 (0.81, 0.96)	1 (Reference)	1 (Reference)	1 (Reference)
	Heavy	77,300	236	487,076.76	0.48	0.37 (0.32, 0.42)	0.80 (0.70, 0.91)	0.83 (0.72, 0.94)	0.90 (0.77, 1.03)	0.93 (0.80, 1.08)	0.93 (0.80, 1.09)
Heavy	None	26,535	106	164,722.03	0.64	0.49 (0.40, 0.59)	1.01 (0.84, 1.23)	1.00 (0.83, 1.22)	1.28 (1.04, 1.58)	1.14 (0.92, 1.42)	1.10 (0.88, 1.38)
(≥30 g/day)	Light	51,744	166	325,094.71	0.51	0.39 (0.33, 0.45)	0.79 (0.68, 0.92)	0.81 (0.69, 0.95)	1.02 (0.85, 1.21)	0.90 (0.75, 1.08)	0.90 (0.75, 1.08)
	Moderate	85,976	265	540,585.87	0.49	0.37 (0.33, 0.42)	0.81 (0.72, 0.92)	0.84 (0.74, 0.95)	0.98 (0.84, 1.14)	0.93 (0.80, 1.08)	0.93 (0.80, 1.09)
	Heavy	146,003	461	917,300.52	0.50	0.38 (0.35, 0.42)	0.88 (0.80, 0.97)	0.90 (0.82, 1.00)	1 (Reference)	1 (Reference)	1 (Reference.)

Incidence rate: per 1000 person-years. * Multivariate adjusted for: age, sex, regular exercise, monthly income, BMI, cigarette smoking, diabetes mellitus, and dyslipidemia. BMI, body mass index; HR, hazard ratio; CI, 95% confidence interval.

**Table 4 cancers-13-02343-t004:** Changes in smoking and alcohol intake on thyroid cancer risk.

Smoking/Drinking Status		Total	Incidence	Person-Years	Incidence Rate	Sustained Non-Smokers/Non-Drinkers as the Reference			Smoking/Drinking Status at the First Examination as the Reference		
At Initial	At 2-Year Follow-Up					Crude HR (95% CI)	Age and Sex Adjusted HR (95% CI)	Multivariate Adjusted * HR (95% CI)	Crude HR (95% CI)	Age and Sex Adjusted HR (95% CI)	Multivariate Adjusted * HR (95% CI)
Non/Non	Non/Non	1,785,110	16,868	11,324,959.38	1.49	1 (Reference)	1 (Reference)	1 (Reference)	1 (Reference)	1 (Reference)	1 (Reference)
	Ever/Non	66,436	256	412,878.93	0.62	0.42 (0.37, 0.47)	0.96 (0.84, 1.09)	0.98 (0.86, 1.11)	0.42 (0.37, 0.47)	0.96 (0.84, 1.09)	0.97 (0.85, 1.10)
	Non/Ever	198,752	1867	1,259,568.95	1.48	0.99 (0.95, 1.04)	0.98 (0.93, 1.02)	0.98 (0.93, 1.03)	0.99 (0.95, 1.04)	0.97 (0.92, 1.02)	0.97 (0.92, 1.01)
	Ever/Ever	57,537	199	362,044.38	0.55	0.37 (0.32, 0.42)	0.82 (0.71, 0.94)	0.82 (0.71, 0.94)	0.37 (0.32, 0.42)	0.81 (0.70, 0.94)	0.80 (0.69, 0.93)
Ever/Non	Non/Non	62,027	215	385,074.89	0.56	0.37 (0.33, 0.43)	0.79 (0.69, 0.91)	0.81 (0.71, 0.93)	1.09 (0.94, 1.27)	0.96 (0.83, 1.12)	0.96 (0.82, 1.12)
	Ever/Non	279,201	891	1,743,937.22	0.51	0.34 (0.32, 0.37)	0.81 (0.75, 0.87)	0.83 (0.77, 0.90)	1 (Reference)	1 (Reference)	1 (Reference)
	Non/Ever	10,003	33	62,762.42	0.53	0.35 (0.25, 0.50)	0.74 (0.53, 1.05)	0.76 (0.54, 1.07)	1.03 (0.73, 1.46)	0.92 (0.65, 1.30)	0.92 (0.65, 1.31)
	Ever/Ever	91,158	284	573,618.11	0.50	0.33 (0.30, 0.37)	0.76 (0.67, 0.86)	0.77 (0.68, 0.87)	0.97 (0.85, 1.11)	0.95 (0.83, 1.09)	0.93 (0.81, 1.06)
Non/Ever	Non/Non	205,146	1795	1,301,114.31	1.38	0.93 (0.88, 0.97)	0.94 (0.90, 0.99)	0.94 (0.90, 0.99)	1.20 (1.13, 1.27)	1.05 (0.99, 1.11)	1.05 (0.99, 1.11)
	Ever/Non	12,887	47	80,215.72	0.59	0.39 (0.29, 0.52)	0.94 (0.70, 1.25)	0.96 (0.72, 1.28)	0.51 (0.38, 0.68)	1.02 (0.77, 1.37)	1.05 (0.79, 1.40)
	Non/Ever	429,147	3140	2,719,020.09	1.15	0.77 (0.75, 0.80)	0.89 (0.86, 0.93)	0.89 (0.86, 0.93)	1 (Reference)	1 (Reference)	1 (Reference)
	Ever/Ever	82,773	281	522,052.53	0.54	0.36 (0.32, 0.41)	0.84 (0.75, 0.95)	0.85 (0.75, 0.96)	0.47 (0.41, 0.53)	0.93 (0.82, 1.06)	0.94 (0.83, 1.07)
Ever/Ever	Non/Non	48,008	171	300,505.18	0.57	0.38 (0.33, 0.44)	0.83 (0.71, 0.97)	0.84 (0.72, 0.98)	1.20 (1.03, 1.40)	1.11 (0.95, 1.30)	1.10 (0.94,1.29)
	Ever/Non	103,047	358	644,784.76	0.56	0.37 (0.33, 0.41)	0.87 (0.78, 0.98)	0.89 (0.79, 0.99)	1.17 (1.05, 1.31)	1.19 (1.06, 1.32)	1.20 (1.07, 1.34)
	Non/Ever	80,098	306	504,629.93	0.61	0.41 (0.36, 0.46)	0.90 (0.80, 1.01)	0.90 (0.80, 1.01)	1.28 (1.14, 1.44)	1.21 (1.07, 1.36)	1.20 (1.07, 1.36)
	Ever/Ever	918,740	2736	5,786,093.86	0.47	0.32 (0.30, 0.33)	0.73 (0.69, 0.77)	0.72 (0.68, 0.77)	1 (Reference)	1 (Reference)	1 (Reference)

Incidence rate: per 1000 person-years. * Multivariate adjusted for: age, sex, regular exercise, monthly income, BMI, diabetes mellitus and dyslipidemia. BMI, body mass index; HR, hazard ratio; CI, 95% confidence interval.

## Data Availability

The datasets used for the current study are available from the corresponding author on reasonable request.

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
