# Peer review of "Changes in Smoking, Alcohol Consumption, and the Risk of Thyroid Cancer: A Population-Based Korean Cohort Study"

_cancers, 2021, doi:10.3390/cancers13102343_

Round 1

Reviewer 1 Report

First I would like to thank for the opportunity to read this very interesting study. The study is rigorously designed with a clear exposition of the criteria and modalities of the sample recruitment. The methods and statistical analysis are correct and clearly stated. The submitted study is a population study, i.e. the Korean one, which because of the large sample and the quality of the selection criteria, must be taken into serious consideration for the purposes of the growing evidence relating to the association between smoking, alcohol consumption and cancers of the Thyroid gland.

I would like to express some personal considerations and suggestions.

The proposed Cox models would need to assess the relative proportional hazard assumptions by means of a formal test.

The tables should be improved in editing.

I agree with the time lag to take into account the "Sick quitter effect" which in this context would produce presumable effects of underestimating the hazard ratios.

The mean follow-up time is much longer than the time between the self-reported survery on smoking and alcohol consumption habits, this can obviously imply further changes in behaviors that cannot be considered in the proposed design. It would be interesting to know the mean times to event (thyroid cancer) from the second self-report, referring only to the group which experienced this event, to provide further information on the importance of a change in the habits.

With reference to section 4.4 (line 273-282), it should be better pointed out that drinking and smoking habits are often statistically associated, raising collinearity issues that can negatively affect the power of statistical analysis.

I do not deny that an analysis that differentiated different levels of severity of thyroid cancer would have further enriched the work. But this may be the subject of new work using, for example, competing risk models.

Author Response

[Reviewer 1]

Comments and Suggestions for Authors

First I would like to thank for the opportunity to read this very interesting study. The study is rigorously designed with a clear exposition of the criteria and modalities of the sample recruitment. The methods and statistical analysis are correct and clearly stated. The submitted study is a population study, i.e. the Korean one, which because of the large sample and the quality of the selection criteria, must be taken into serious consideration for the purposes of the growing evidence relating to the association between smoking, alcohol consumption and cancers of the Thyroid gland.

I would like to express some personal considerations and suggestions.

Comment #1) The proposed Cox models would need to assess the relative proportional hazard assumptions by means of a formal test.

Response) The proportional hazard assumption was well preserved with the Schonfeld residuals with the cumulative hazard functions in the Kaplan-Meier curves. We further described this in the Methods section (Page 6, 1st Paragraph).

“The proportional hazards assumption was checked by using the Schoenfeld residuals with the logarithm of the cumulative hazards function estimated by the Kaplan–Meier method.”

Comment #2) The tables should be improved in editing.

Response) We appreciate reviewer’s comment. We edited the tables to improve readability.

Comment #3) I agree with the time lag to take into account the "Sick quitter effect" which in this context would produce presumable effects of underestimating the hazard ratios.

Response) As the reviewer’s comment, the sick quitter effect should be considered in the study of those who modified their lifestyle, such as smoking, alcohol consumption or physical activities. The onset of a disease or changes in health conditions can be lead to cessation of smoking or drinking, and this effect usually leads to an underestimation of the association (an overestimation of the inverse association in this study) between lifestyle changes and outcome diseases.

Theses quitters might have higher probability to develop the diseases that was not detected at the time of study enrollment in longitudinal studies. In our study, those who were not diagnosed with thyroid cancer (disease of outcome) or certain comorbidities that are related to thyroid cancer diagnosis (e.g. thyroid cancer is often diagnosed simultaneously through health check-up which is performed for detecting other diseases), but had self-perception of poor health conditions might have quitted smoking or drinking alcohol before the manifestation of the disease. To avoid this, measures such as specifying the reasons for quitting, excluding former users (former drinkers or smokers and lifetime abstainers) from the study population, or applying a lag-period for the occurrence of diseases are taken in the studies about lifestyle changes and health outcome.

As the reviewer pointed out, applying a time lag to consider the “sick quitter effect” could have risk for presumable effects of underestimating the hazard ratios. However, for the above-mentioned reason, we decided to apply the lag-period for cancer, cardiovascular diseases, and deaths to wash-out the possibility of poor health conditions that could have led to smoking cessation after the study enrollment (date of second health examination). The results remained significant even after application of the lag-time.

Comment #4) The mean follow-up time is much longer than the time between the self-reported survery on smoking and alcohol consumption habits, this can obviously imply further changes in behaviors that cannot be considered in the proposed design. It would be interesting to know the mean times to event (thyroid cancer) from the second self-report, referring only to the group which experienced this event, to provide further information on the importance of a change in the habits.

Response) As the reviewer commented, the mea time to event (thyroid cancer) from the second self-report refers only to the group which experienced this event, and it would provide further information on the importance of a change in the habits. Therefore, we added the mean time to thyroid cancer incidence in the Results section (Page 6, 2nd Paragraph)

“During the mean follow-up period of 6.32±0.72 years, 29,447 individuals were diagnosed with thyroid cancer. The mean time to thyroid cancer incidence was 2.66 ± 1.88 years.

Smoking and drinking behaviors can constantly change. However, we measured these behaviors only only at two points, with a two-year interval. Therefore, as the reviewer pointed out, the mean follow-up time (e.g. 6.3 -years ) is much longer than the period between self-reported survey on smoking and alcohol consumption habits (i.e. two years) and further changes in behaviors after the second examination could not be properly reflected in this study. We added this as limitation in the Discussion section (Page 12, 4.6 Limitation).

“Fifth, while smoking and drinking behaviors can constantly change, we measured these behaviors only at two timepoints, and did not reflect changes that were made in the rest of the follow-up period.

Comment #5) With reference to section 4.4 (line 273-282), it should be better pointed out that drinking and smoking habits are often statistically associated, raising collinearity issues that can negatively affect the power of statistical analysis.

Response) We agree with reviewer’s comment that smoking and drinking habits are often correlated. As the reviwer pointed out, the joint effect between smoking and alcohol consumption should be regarded to explore the associations, as the collinearity issues can affect the results of statistical analysis.

Therefore, we estimated the joint effect of the behaviros (changes in smoking x changes in alcohol intake) on the incidence of thyroid cancer (describe in Table 4), and also included each behavior as a covariate (i.e. adjusting alcohol intake for the analysis of smoking, and vice versa) in the multivariable models (described in Table 2 and Table 3). We further described this in the Discussion section (Page 12, 4.5 Joint effect of changes in smoking and alcohol intake).

“As smoking and drinking habits are often correlated and would have collinearity issues, we estimated their joint effect.”

Comment #6) I do not deny that an analysis that differentiated different levels of severity of thyroid cancer would have further enriched the work. But this may be the subject of new work using, for example, competing risk models.

Response) Agreeing with reviewer’s comment, the histologic type or stage of thyroid cancer might be important to differentiate the severity or the characteristics of thyroid cancer. Unfortunately, from the KNHI database, which our study was based on, the information on the histologic type or the stage of thyroid cancers was not available. We thought that future studies regarding the difference in the severity of thyroid cancer should be followed.

Reviewer 2 Report

In their study, Yeo and co-workers aimed to explore the effect(s) of changes in smoking and alcohol consumption on thyroid cancer incidence.

The study is very large (29,447 patients with thyroid cancer), and well designed. The conclusion are clearly stated, and the limits of the study detailed by the Authors.

The results support evidence that smoking and (possibly) alcohol consumption may have a protective effect, being the analysis of changes in both smoking and alcohol consumption the most intriguing part.

Minor comments: there is a clerical error in the number of identified participants: 4,961,411 in Figure 1, and 4,961,441 in the Methods Section (page 3, line 105).

Author Response

[Reviewer 2]

Comments and Suggestions for Authors

In their study, Yeo and co-workers aimed to explore the effect(s) of changes in smoking and alcohol consumption on thyroid cancer incidence.

The study is very large (29,447 patients with thyroid cancer), and well designed. The conclusion are clearly stated, and the limits of the study detailed by the Authors.

The results support evidence that smoking and (possibly) alcohol consumption may have a protective effect, being the analysis of changes in both smoking and alcohol consumption the most intriguing part.

Comment #1) Minor comments: there is a clerical error in the number of identified participants: 4,961,411 in Figure 1, and 4,961,441 in the Methods Section (page 3, line 105).

Response) Thank you for your meticulous examination. We revised this error clearly in Figure 1.
